# EPO-R76E Enhances Retinal Pigment Epithelium Viability Under Mitochondrial Oxidative Stress Induced by Paraquat

**DOI:** 10.3390/cells14221794

**Published:** 2025-11-14

**Authors:** Jemima Alam, Alekhya Ponnam, Arusmita Souvangini, Sundaramoorthy Gopi, Cristhian J. Ildefonso, Manas R. Biswal

**Affiliations:** 1Department of Pharmaceutical Sciences, Taneja College of Pharmacy, University of South Florida, Tampa, FL 33620, USAalekhya12@usf.edu (A.P.); arusmita@usf.edu (A.S.); gopis@usf.edu (S.G.); 2Department of Ophthalmology, College of Medicine, University of Florida Gainesville, Gainesville, FL 32610, USA; 3Department of Ophthalmology, Morsani College of Medicine, University of South Florida, Tampa, FL 33620, USA

**Keywords:** retinal degeneration, mitochondrial dysfunction, erythropoietin, ARPE-19, paraquat, RPE, antioxidants, growth factors

## Abstract

Age-related macular degeneration (AMD) is a leading cause of irreversible vision loss, primarily driven by oxidative stress–induced degeneration of retinal pigment epithelium (RPE). Erythropoietin (EPO), a hematopoietic cytokine with neuroprotective properties, has been shown to reduce apoptosis and retinal degeneration. In this study, we examined the cytoprotective role of a non-erythropoietic EPO variant, EPO-R76E, in suppressing oxidative stress and mitochondrial dysfunction related to oxidative stress in RPE cells. Stable ARPE-19 cell lines expressing EPO-R76E were generated via lentiviral transduction and exposed to paraquat to induce oxidative stress. Oxidative stress was induced using paraquat. EPO-R76E expression conferred increased cell viability and resistance to mitochondrial damage, as assessed by cytotoxicity assays. Western blot analysis revealed reduced expression of ferritin and p62/SQSTM1, diminished activation of p-AMPK and NRF2, and restoration of GPX4 levels, indicating enhanced antioxidant defenses. Moreover, intracellular iron accumulation and reactive oxygen species were significantly reduced in EPO-R76E-expressing cells exposed to paraquat. These findings suggest that EPO-R76E promotes mitochondrial homeostasis and modulates oxidative stress pathways. Our study positions EPO-R76E as a promising therapeutic candidate for halting RPE degeneration in AMD.

## 1. Introduction

Age-related macular degeneration (AMD) is a leading cause of irreversible vision impairment in the aging population, characterized by progressive degeneration of the macula affecting the retinal pigment epithelium (RPE) and photoreceptors [1]. Among its two subtypes, neovascular AMD (nAMD) has benefited from anti-VEGF therapies, while dry AMD (dAMD), accounting for over 90% of cases, still lacks effective treatment options due to elusive pathogenesis [2]. A growing body of evidence links dAMD to oxidative stress and mitochondrial dysfunction in RPE cells, which play a critical role in nutrient transport, visual cycle maintenance, phagocytosis, and cellular waste clearance [3,4,5]. The mechanism of oxidative stress-induced damage involves several pathways.

Oxidative stress arises from an imbalance between reactive oxygen species (ROS) production and antioxidant defenses, often exacerbated by aging, light exposure, and iron accumulation [6,7]. Mitochondria, the primary source of endogenous ROS, become both targets and amplifiers of oxidative damage, leading to a vicious cycle of mitochondrial dysfunction and RPE cell death [8,9]. This stress-induced dysfunction has been associated with a variety of distinct cell death pathways, including apoptosis, necroptosis, pyroptosis, and ferroptosis, depending on the specific cellular stressor or context [10]. Ferroptosis, a regulated, iron-dependent form of cell death characterized by lipid peroxidation and impaired antioxidant defenses such as glutathione (GSH) and glutathione peroxidase 4 (GPX4) can act as a potential target to mitigate dAMD [11]. In response to the dysfunctional mitochondria triggered by ferroptosis and oxidative stress, cells employ sophisticated defense systems.

EPO-R76E, a non-erythropoietic variant of erythropoietin, has emerged as a candidate neuroprotective agent. Retaining anti-oxidative and anti-apoptotic properties, EPO-R76E has shown promise in retinal models [12,13,14]. We have also shown that EPO-R76E protects RPE from Sod2 deletion induced oxidative damage [12]. However, the ability of EPO-R76E to protect retinal pigment epithelial (RPE) cells from mitochondrial oxidative stress remains unexplored. Paraquat-induced oxidative stress in RPE cells is widely used as an in vitro model of dry age-related macular degeneration (AMD) because it reproduces key cellular features observed in the disease. Paraquat, a redox-cycling herbicide, generates mitochondrial reactive oxygen species (ROS), leading to mitochondrial dysfunction, ATP depletion, and energy stress, which are hallmarks of RPE injury in dry AMD. This stress activates AMPK phosphorylation and stabilizes NRF2, triggering antioxidant defenses, thereby recapitulating the compensatory stress responses observed in the diseased retina. Furthermore, paraquat can induce ferroptosis-like, iron-dependent lipid peroxidation, a mechanism increasingly implicated in dry AMD pathogenesis. Together, these effects make paraquat a relevant and reproducible model for investigating the cellular mechanisms of RPE injury and testing protective interventions such as EPO-R76E. We hypothesize that sustained expression of EPO-R76E preserves mitochondrial integrity, thereby maintaining ATP levels, reducing ROS accumulation, and consequently attenuating both p-AMPK activation and NRF2 signaling (Figure 1).

To test this, we employed ARPE-19 cells stably expressing lentivirally delivered EPO-R76E and evaluated their response to oxidative stress. This study aims to define the cytoprotective role of EPO-R76E and provide mechanistic insight into its therapeutic potential for preventing RPE degeneration in dry age-related macular degeneration (dAMD).

## 2. Materials and Methods

### 2.1. Plasmid Preparation

The process of preparing the Lenti-EPO-R76E plasmid involved several sequential molecular biology techniques aimed at cloning and validating the construct within a lentiviral vector backbone.

#### 2.1.1. Gradient PCR Amplification of EPO-R76E Insert

The procedure began with the amplification of the EPO-R76E gene (provided by Dr. Tonia S. Rex from Vanderbilt University) using a gradient PCR (Bio-Rad T100 Thermal cycler, Hercules, CA, USA) to optimize annealing temperatures. The EPO-R76E plasmid served as the template, and amplification was performed using EcoR1 forward primer (Integrated DNA Technologies. Coralville, IA, USA Cat no: 470352456), Not1 reverse primer (Integrated DNA Technologies, Coralville, IA, USA Cat no: 470352456), and Promega GoTaq™ G2 Green Master Mix (Fisher Scientific, Waltham, MA, USA, Cat no: PRM7823). The reaction mixture was subjected to Gradient PCR (Bio-Rad, S Hercules, California) at 96 °C for 6 min, 61–69 °C for 1 min, and 72 °C for 10 min.

#### 2.1.2. Gel Electrophoresis and DNA Elution

The amplified product was analyzed by agarose gel electrophoresis. A small gel was prepared with 1% agarose gel with six wells. After electrophoresis, the DNA bands were visualized under a UV illuminator (Bio-rad Gel Doc XR+Molecular Imager, Hercules, CA, USA), and the expected band corresponding to the EPO-R76E insert was carefully excised. The DNA fragment was then eluted using MinElute gel extraction kit (Qiagen, Germantown, MD, USA, Cat no: 28606) according to the manufacturer’s instructions.

#### 2.1.3. Quantification and Restriction Digestion

The eluted PCR product was quantified using a Nanodrop spectrophotometer (Thermo Fisher Scientific, Madison, WI, USA) to ensure sufficient DNA concentration and purity for downstream applications. Both the EPO-R76E insert, and the lentiviral vector plasmid backbone were subjected to restriction digestion using FastDigest EcoRI (Thermo Fisher Scientific, Waltham, MA, USA, Cat no: FD0274) and NotI (Thermo Fisher Scientific, Waltham, MA, USA, Cat no: FD0593) restriction enzymes. The digested products were run again on a 1% agarose gel to confirm successful digestion. The relevant DNA bands were excised, eluted, and quantified as previously described.

#### 2.1.4. Ligation and Bacterial Transformation

A ligation reaction was set up using the T4 DNA ligase enzyme (Thermo Fisher Scientific, Cat no: EL0011) to ligate the digested EPO-R76E insert into the similarly digested lentiviral vector backbone. The ligation mixture was then transformed into Max Efficiency DH5α competent cells (Invitrogen, Carlsbad, CA, USA. Cat no: 18258012) using the heat shock method which include keeping on ice for 15 min, heating at 42 °C for 45 s, and again keeping on ice for 3 min. S.O.C medium (Invitrogen, Cat no: 15544034) was added to the cells and kept on continuous shaking for 45 min. After 45 min, the solution turned cloudy, and it was centrifuges for 6 min and the supernatant was discarded. The pellets were resuspended in fresh S.O.C medium. The transformed cells were plated onto LB agar plates containing the appropriate antibiotic selection marker and incubated overnight at 37 °C.

#### 2.1.5. Colony Screening and Plasmid Isolation

Following incubation, colonies presumed to contain the Lenti-EPO-R76E construct were observed. Individual colonies were picked and inoculated into LB broth for overnight culture. Miniprep plasmid isolation was performed using a commercial plasmid extraction kit (GeneJET Plasmid Miniprep Kit, Thermo Scientific, Vilniun, Lithuania). The isolated plasmid DNA was again quantified using a Nanodrop spectrophotometer (Thermo Fisher Scientific, Madison, WI, USA).

#### 2.1.6. Confirmation by Restriction Analysis and In Silico Validation

To confirm successful cloning, the isolated plasmid DNA was subjected to a second round of restriction digestion using FastDigest EcoRI (Thermo Fisher Scientific, Cat no: FD0274) and NotI (Thermo Fisher Scientific, Cat no: FD0593) enzymes. The digested products were run on an agarose gel to verify the presence and size of the inserted gene. The resulting banding pattern was compared to expected fragment sizes using APE (A Plasmid Editor) software for in silico validation, confirming the successful construction of the Lenti-EPO-R76E plasmid. The results are provided in the Appendix A.

#### 2.1.7. Validation of Lenti-EPO-R76E Plasmid

To confirm the successful construction and functionality of the Lenti-EPO-R76E plasmid, HeLa cells were transfected and analyzed using fluorescence microscopy and Western blotting.

HeLa cells were seeded in a 6-well plate at a density of 50,000 cells per well in 2 mL of complete EMEM (ATCC, Manassas, VA, USA, Cat no: 30-2003) (supplemented with 10% fetal bovine serum (FBS) (ATCC, Manassas, VA, USA, Cat no: 30-2020) and 1% penicillin-streptomycin (Gibco, Grand Island, NY, USA, Cat no: 15070-063)) medium and incubated overnight at 37 °C with 5% CO_2_ to allow for cell attachment. On the following day, transfection was carried out using polyethylenimine (PEI) (Polysciences, Warrington, PA, Cat no: 23966-100) as the transfection reagent. For each well, Lenti-EPO-R76E or GFP plasmid DNA was mixed with PEI at a 1:1 ratio in Opti-MEM (Gibco, Cat no: 31985070) and incubated at room temperature for 30 min to allow complex formation. The transfection mixture was then added dropwise to the corresponding wells: three wells received Lenti-EPO-R76E plasmid, two wells received a GFP-expressing plasmid to evaluate transfection efficiency, one well served as an un-transfected negative control. Cells were incubated for 24 h post-transfection under standard culture conditions.

After 24 h, GFP-transfected cells were imaged using a fluorescence microscope to confirm the efficiency of transfection. Random fields were captured under identical settings to ensure consistency across samples.

Post-transfection, cells were washed with cold phosphate-buffered saline (PBS) (Sigma, Lenexa, KS, USA, Cat no: D8537) and lysed using RIPA lysis buffer (Thermo Fisher Scientific, Cat no: 89901) containing protease inhibitors (Sigma-Aldrich, Cat no: P8340). Protein concentration was determined using the Pierce™ 660 nm Protein Assay (Thermo Fisher Scientific, Cat no: 22660) using Pre-diluted Protein Assay Standard BSA set (Thermo Fisher Scientific, Cat no: 23208) as a standard.

20 μg of protein from each sample were subjected to SDS-PAGE (Invitrogen, Cat no: NP0321BOX) and transferred onto PVDF membranes (Invitrogen, Cat no: IB23002). The membranes were blocked with Intercept (PBS) blocking buffer (LI-COR Biosciences, Lincoln, NE, USA, Cat no: 927-70001) for an hour and probed overnight with primary antibodies anti-2A peptide rabbit (Sigma-Aldrich, Cat no: ABS31-1) with 1:2000 dilution and β-actin mouse (Santa Cruz Biotechnology, Dallas, TX, USA, Cat no: SC-47778) with 1:2000 dilution. The anti-2A peptide antibody was used to specifically detect EPO-R76E expression, as the LentiEPO-R76E construct includes a T2A self-cleaving peptide sequence. The next day, the membrane was washed four times with washing buffer containing PBS-Tween 20 (0.1%), followed by incubation with appropriate secondary antibodies goat anti mouse (LI-COR Biosciences, Lincoln, NE, USA, Cat no: 926-32210) and donkey anti rabbit (LI-COR Biosciences, Cat no: 926-68073) with 1:5000 dilution for 45 min. The membrane was again washed four times with the washing buffer and then the signal detection was carried out using Licor image studio software. The results are provided in the Appendix A.

### 2.2. Lentivirus Production

The production of recombinant lentivirus expressing EPO-R76E was carried out using transient transfection of HEK293T cells with the plasmid construct and essential packaging plasmids.

#### 2.2.1. Cell Seeding for Lentivirus Production

HEK293T cells, known for their high transfection efficiency, were seeded into a 6-well tissue culture plate at an appropriate density to achieve approximately 70–80% confluency on the day of transfection. The cells were maintained in complete DMEM (Gibco, Cat no: 10313039) supplemented with 10% FBS (ATCC, Cat no: 30-2020) and 1% penicillin-streptomycin (Gibco, Cat no: 15070-063).

#### 2.2.2. Plasmid Transfection

For lentiviral particle production, the cells were co-transfected with Lenti-EPO-R76E (expression plasmid), pRSV-Rev (rev expression plasmid), pMDL (packaging plasmid), and pMD2.G (VSV-G envelope plasmid). Transfection was performed using PEI (Polysciences, Cat no: 23966-100) as the transfection reagent. The plasmid DNA and PEI were mixed in a 1:1 ratio (*w*/*w*) in Opti-MEM (Gibco, Cat no: 31985070) incubated at room temperature for 15–20 min to allow complex formation and then added dropwise to the cells. Plates were gently swirled to ensure even distribution, and cells were incubated at 37 °C in a 5% CO_2_ incubator for 48 h.

#### 2.2.3. Virus Harvesting and Purification

After 48 h of incubation, the culture medium containing the lentiviral particles was collected. The medium was subjected to low-speed centrifugation to remove cellular debris. Then the viral particles were concentrated using PEG-it, a commercial viral concentration solution (System Biosciences, Palo Alto, CA, USA, Cat no: LV810a-1) and stored at 4 °C overnight. The next day, it was centrifuged at 1500× *g* for 30 min. The supernatant was discarded, and the pellets were collected and resuspended in serum free DMEM. The purified Lenti-EPO-R76E viral stock was aliquoted and stored at −80 °C until further use in downstream applications.

### 2.3. Stable ARPE-19 Cell Line Generation

The generation of a stable ARPE-19 cell line expressing the EPO-R76E transgene was achieved via lentiviral transduction followed by antibiotic selection.

#### 2.3.1. Cell Preparation and Transduction

ARPE-19 cells were seeded in T-25 flask to achieve approximately 60–70% confluency at the time of transduction. The cells were maintained in DMEM/F12 (ATCC, Cat no: 30-2006) medium supplemented with 10% FBS (ATCC, Cat no: 30-2020) and 1% penicillin-streptomycin (Gibco, Cat no: 15070-063)

Lentiviral particles carrying the Lenti-EPO-R76E construct were thawed on ice and added to the ARPE-19 cells in the presence of 10 μg/mL polybrene (Millipore Sigma, Cat no: TR-1003), a cationic polymer used to enhance viral transduction efficiency. The transduction mixture was incubated with the cells for 24 h at 37 °C in a 5% CO_2_ atmosphere, after which the medium was replaced with fresh complete growth medium.

#### 2.3.2. Selection of Stable Clones

To ensure the establishment of a stably transduced cell population, transduced ARPE-19 cells were subjected to puromycin (Sigma-Aldrich, Cat no: P8833) selection. A puromycin kill curve had been previously optimized to determine the minimum concentration required to eliminate non-transduced cells. Based on this, puromycin was added at the optimized concentration (10 μg/mL) to the culture medium, and cells were monitored for survival.

Only those cells that had successfully integrated the Lenti-EPO-R76E construct, conferring puromycin resistance, survived the antibiotic selection. The medium was replenished every alternate day and puromycin selection was maintained for approximately 7 days until a pure population of resistant cells was established. The establishment of the stable ARPE-19-EPO-R76E cell line was confirmed by the survival of cells under continuous puromycin selection.

#### 2.3.3. Validation of Stable EPO-R76E Expression in ARPE-19 Cells

Stable expression of EPO-R76E in ARPE-19 cells was confirmed by Western blot. Stably transduced (ARPE-19-EPO-R76E) and control cells were cultured to ~80% confluence, washed with ice-cold PBS (Sigma-Aldrich, Cat. No: D8537), and lysed in RIPA buffer (Thermo Fisher Scientific, Cat. No: 89901) supplemented with protease inhibitor cocktail (Sigma-Aldrich, Cat. No: P8340). Protein concentrations were determined using the Pierce™ 660 nm Protein Assay (Thermo Fisher Scientific, Cat. No: 22660) with a pre-diluted BSA standard (Thermo Fisher Scientific, Cat. No: 23208).

Equal amounts of protein (20 µg) were separated on NuPAGE 4–12% Bis-Tris gels (Invitrogen, Cat. No: NP0321BOX) and transferred to PVDF membranes (Invitrogen, Cat. No: IB23002). Membranes were blocked in Intercept (PBS) blocking buffer (LI-COR, Cat. No: 927-70001) for 1 h at room temperature and incubated overnight at 4 °C with primary antibodies diluted in blocking buffer: Epo polyclonal antibody (Invitrogen, Cat. No: PA5-79212, 0.5 µg/mL) and β-actin (Santa Cruz Biotechnology, Cat. No: SC-47778, 1:2000).

After four washes with PBS containing 0.1% Tween-20 (PBS-T), membranes were incubated with fluorescent secondary antibodies (goat anti-mouse IgG, LI-COR, Cat. No: 926-32210; donkey anti-rabbit IgG, LI-COR, Cat. No: 926-68073) diluted 1:5000 in blocking buffer for 45 min at room temperature. Membranes were washed four additional times with PBS-T, and signals were detected using the LI-COR Odyssey imaging system and analyzed with LI-COR Image Studio software (Version 5.5.4). The results are shown in Appendix A.

### 2.4. Cell Viability Assay

A comparative cell viability assay was conducted to assess the protective effect of EPO-R76E against oxidative stress in ARPE-19 and stable Lenti-EPO-R76E-transduced ARPE-19 cells, using the WST-1 assay following exposure to paraquat-induced cytotoxicity.

#### 2.4.1. Cell Seeding for Cell Viability Assay

ARPE-19 cells and stable Lenti-EPO-R76E-expressing ARPE-19 cells were separately seeded into 96-well plates at a density of 10,000 cells per well in 100 µL of complete culture medium. The plates were incubated overnight at 37 °C in a 5% CO_2_ incubator to allow for cell attachment and recovery.

#### 2.4.2. Paraquat Treatment

The following day, the culture medium was aspirated and replaced with fresh serum-free medium containing paraquat (Sigma-Aldrich, Cat no: 102677317) at varying concentrations: 0 µM (control), 200 µM, 400 µM, 600 µM, 800 µM, and 1000 µM. Each condition was added to five wells to ensure statistical accuracy. The cells were incubated with paraquat for 24 h under standard culture conditions to induce oxidative stress.

#### 2.4.3. WST-1 Assay for Viability Measurement

Following the 24 h treatment period, WST-1 reagent (Roche Diagnostics GmbH, Mannheim, Germany, Cat no: 11644807001) was added to each well as per the manufacturer’s protocol (10 µL per 100 µL of culture medium). The plates were incubated for an additional 3 h at 37 °C to allow for the formation of formazan dye, which is directly proportional to the number of viable cells.

#### 2.4.4. Absorbance Measurement

After incubation, the absorbance was measured at 450 nm using a microplate reader (BioTek Synergy, Winooski, VT, USA). The resulting absorbance values were used to quantify cell viability across different paraquat concentrations in both control and EPO-R76E-expressing ARPE-19 cells.

#### 2.4.5. Data Analysis

Data were analyzed using GraphPad Prism (Version 10.6.0) to assess the dose-dependent cytotoxic effect of paraquat and the potential protective role of EPO-R76E against oxidative stress-induced cell death. All experiments were performed in triplicate to ensure reproducibility and statistical validity.

### 2.5. Protein Expression Analysis

Western blot analysis was performed to assess the expression of key proteins involved in autophagy and oxidative stress response in ARPE-19 and Lenti-EPO-R76E-transduced ARPE-19 cells. The analysis included detection of ferritin, p62, GPX4, and phosphorylated AMPK (p-AMPK). β-actin was used as an internal loading control to normalize protein expression levels.

#### 2.5.1. Cell Culture and Protein Extraction

ARPE-19 cells and stable Lenti-EPO-R76E-expressing ARPE-19 cells were separately seeded into 6-well plates at a density of 50,000 cells per well in 2 mL of complete culture medium (DMEM/F12 supplemented with 10% FBS and 1% penicillin-streptomycin). The plates were incubated overnight at 37 °C in a 5% CO_2_ incubator to allow for cell attachment and recovery.

To induce mitochondrial oxidative stress, ARPE-19 cells were treated with 400 µM paraquat (Sigma-Aldrich, Cat. No: 102677317) for 24 h [15,16]. Control cells received incomplete culture medium (ICM) without paraquat. Each condition was prepared in triplicate (*n* = 3) to ensure reproducibility. After treatment, cells were washed with ice-cold PBS (Sigma-Aldrich, Cat. No: D8537) and lysed using RIPA lysis buffer (Thermo Fisher Scientific, Cat. No: 89901) supplemented with protease inhibitor cocktail (Sigma-Aldrich, Cat. No: P8340) and phosphatase inhibitor cocktail (Thermo Fisher Scientific, Cat. No: 78428). Lysates were incubated on ice for 30 min and centrifuged at 14,000× *g* for 10 min at 4 °C. The resulting supernatants were collected and stored at −80 °C until use.

#### 2.5.2. Protein Quantification

Protein concentration was determined using the Pierce™ 660 nm Protein Assay Reagent (Thermo Fisher Scientific, Cat. No: 22660), with pre-diluted BSA standards (Thermo Fisher Scientific, Cat. No: 23208) as a reference. A standard curve was generated using serial dilutions of BSA, and absorbance was measured at 660 nm using a microplate reader (BioTek Synergy, Winooski, VT, USA). Protein concentrations were calculated from the standard curve, and samples were diluted to equal concentrations for SDS-PAGE. Each sample was mixed with NuPAGE LDS Sample Buffer (4X) (Invitrogen, Cat. No: NP0007) and heated at 70–80 °C for 10 min for denaturation.

#### 2.5.3. Electrophoresis and Transfer

Equal amounts (20 µg) of total protein were resolved using NuPAGE 4–12% Bis-Tris SDS-PAGE gels (Invitrogen, Cat. No: NP0321BOX). Following electrophoresis, proteins were transferred onto nitrocellulose membranes (Invitrogen, Cat. No: IB23001) using the iBlot 2 Dry Blotting System (Invitrogen, Cat. No: IB21001). Membranes were blocked for 1 h at room temperature using Intercept (PBS) Blocking Buffer (LI-COR Biosciences, Cat. No: 927-70001).

#### 2.5.4. Primary Antibody Incubation

Blocked membranes were incubated overnight at 4 °C with the following primary antibodies diluted in blocking buffer according to manufacturer-recommended dilutions:Ferritin (Rabbit monoclonal)—MA5-32244, Invitrogen, 1:5000;SQSTM1/p62 (Rabbit polyclonal)—Abcam, Cat. No: ab109012, 1:10,000;GPX4 (Rabbit monoclonal)—Abcam, Cat. No: ab125066 or Proteintech, Cat. No: 67763-1-Ig, 1:1000;NRF2 polyclonal antibody—PA5-27882, Invitrogen, 1:1000;Phospho-AMPKα (Thr172, Rabbit monoclonal)—Cell Signaling Technology, Cat. No: 2537S, 1:1000;β-Actin (Mouse monoclonal)—Santa Cruz Biotechnology, Cat. No: SC-47778, 1:2000.

#### 2.5.5. Secondary Antibody Incubation and Detection

Following primary antibody incubation, membranes were washed four times with PBS containing 0.1% Tween-20 (PBS-T) and then incubated for 45 min at room temperature with LI-COR IRDye–conjugated secondary antibodies, Goat anti-mouse IgG IRDye^®^ 800CW (Cat. No: 926-32210) and Donkey anti-rabbit IgG IRDye^®^ 680RD (Cat. No: 926-68073), diluted 1:5000 in blocking buffer. After secondary incubation, membranes were washed four additional times with PBS-T, and signals were detected using the LI-COR Odyssey Imaging System.

#### 2.5.6. Densitometric Analysis

Band intensities were quantified using ImageJ software (Version 1.53m). The expression levels of target proteins were normalized against β-actin to correct for loading variability. Relative protein levels were compared between control and experimental groups to assess the biological effects of EPO-R76E expression. The results were analyzed in GraphPad prism. All experiments were performed in triplicate to ensure statistical reliability.

### 2.6. Intracellular Ferrous Ion Content Assay

The intracellular iron levels in ARPE-19 and Lenti-EPO-R76E–expressing ARPE-19 cells were measured using FerroOrange (Cell Signaling Technology, Kumamoto, Japan, Cat. No: 36104S), a fluorescent probe specific for labile Fe^2+^. Fluorescence was captured by microscopy (Keyence BZX800, Osaka, Japan) and quantified using ImageJ (Version 1.53m).

#### 2.6.1. Cell Seeding and Treatment for Intracellular Ferrous Ion Content Assay

Cells were seeded in a 12-well plate with 6 wells for ARPE-19 and 6 wells for Lenti-EPO-R76E-expressing ARPE-19 cells at a density of 50,000 cells per well in complete culture medium (DMEM/F12 supplemented with 10% FBS and 1% penicillin-streptomycin) and incubated overnight at 37 °C in a 5% CO_2_ incubator to allow for cell attachment.

The following day, the culture medium was replaced with serum-free medium containing 400 µM paraquat (Sigma-Aldrich, Cat. No: 102677317) to induce oxidative stress and promote intracellular iron accumulation. Control wells received serum-free medium without paraquat. Cells were incubated under these conditions for 24 h.

#### 2.6.2. FerroOrange Staining

After paraquat treatment, the medium was carefully removed, and cells were washed three times with sterile PBS (Sigma-Aldrich, Cat. No: D8537) to eliminate residual paraquat and serum components. Cells were then incubated with 1 µM FerroOrange (Cell Signaling Technology, Kumamoto, Japan, Cat No: 36104S) diluted in serum-free medium for 30 min at 37 °C in the dark to prevent photobleaching.

#### 2.6.3. Imaging and Quantification of FerroOrange Staining

Following incubation, cells were imaged immediately using a fluorescence microscope for FerroOrange (Cell Signaling Technology, Cat No: 36104S). Multiple random fields were captured per well to ensure representative sampling. The fluorescence intensity, indicative of intracellular Fe^2+^ levels, was quantified using ImageJ software. Mean intensity values were calculated for each field and averaged across replicates. All experiments were performed in triplicate to ensure statistical reliability.

### 2.7. Reactive Oxygen Species Analysis

Intracellular reactive oxygen species (ROS) levels were measured in ARPE-19 and Lenti-EPO-R76E–expressing ARPE-19 cells using the cell-permeable fluorescent dye DCFDA (2′,7′-dichlorofluorescin diacetate, Sigma-Aldrich, Cat. No: D6883). Fluorescence signals were captured using a Keyence BZX-800 microscope and quantified with ImageJ.

#### 2.7.1. Cell Seeding and Treatment for ROS Analysis

Cells were seeded in a 12-well plate with 6 wells for ARPE-19 and 6 wells for Lenti-EPO-R76E-expressing ARPE-19 cells at a density of 50,000 cells per well in complete culture medium and incubated overnight at 37 °C in a 5% CO_2_ incubator to allow for proper attachment. On the following day, the culture medium was replaced with serum-free medium containing 400 µM paraquat (Sigma-Aldrich, Cat. No: 102677317) to induce oxidative stress. Control wells received serum-free medium without paraquat. Cells were incubated under these conditions for 24 h.

#### 2.7.2. DCFDA Staining

After treatment, the medium was aspirated, and cells were washed three times with PBS to remove residual treatment compounds. Cells were then incubated with 10 µM DCFDA (Sigma-Aldrich, Cat. No: D6883) prepared in serum-free medium for 30 min at 37 °C in the dark, allowing the dye to enter the cells and become oxidized by ROS into fluorescent DCF.

#### 2.7.3. Imaging and Quantification of DCFDA Staining

Following staining, cells were washed with PBS and imaged using a fluorescence microscope (Keyence BZ-X800, Osaka, Japan). Multiple random fields per well were captured under identical exposure settings to ensure consistency. Fluorescence intensity, reflecting intracellular ROS levels, was quantified using ImageJ software. Mean intensity values were calculated for each field and averaged across replicates. All experiments were performed in triplicate to ensure reproducibility and statistical validity.

## 3. Results

### 3.1. Cell Viability Under Oxidative Stress

To assess the protective effect of stable EPO-R76E expression against oxidative stress-induced RPE cell death, ARPE-19 cells and ARPE-19 cells stably expressing EPO-R76E were treated with increasing concentrations of paraquat (0, 200, 400, 600, 800, and 1000 µM) for 24 h. Cell viability was determined using the WST-1 assay, measuring absorbance.

In control ARPE-19 cells, paraquat treatment induced a dose-dependent reduction in cell viability. Mean absorbance values were recorded as 1.36 at 0 µM paraquat, decreasing to 1.11 at 200 µM, 0.87 at 400 µM, 0.74 at 600 µM, 0.68 at 800 µM, and 0.61 at 1000 µM paraquat. In contrast, ARPE-19 cells stably expressing EPO-R76E exhibited significantly higher cell viability across all tested paraquat concentrations compared to control cells. Absorbance values for stable EPO-R76E cells were 1.45 at 0 µM, 1.32 at 200 µM, 1.01 at 400 µM, 0.84 at 600 µM, 0.75 at 800 µM, and 0.69 at 1000 µM paraquat (Figure 2).

### 3.2. Impact on Oxidative Stress-Related Proteins by EPO-R76E

To further investigate the mechanisms by which EPO-R76E expression protects RPE cells from oxidative stress-induced mitochondrial damage, we performed Western blot analysis on protein lysates from ARPE-19 cells and stable Lenti-EPO-R76E-expressing ARPE-19 cells treated with 400 µM paraquat for 24 h. We examined the expression levels of key proteins involved in oxidative stress regulation, including ferritin, sqstm1/p62, NRF2, p-AMPK, and GPX4 (Figure 3A).

Western blot analysis of ferritin, an iron-storage protein often upregulated during paraquat-induced oxidative stress induction, revealed differences between the cell lines. Under control conditions (0 µM paraquat), ferritin levels were slightly lower in stable EPO-R76E-expressing ARPE-19 cells compared to ARPE-19 cells. Upon treatment with 400 µM paraquat, ferritin levels increased in both cell types. However, the increase in ARPE-19 cells was significantly greater compared to their control, while the increase in EPO-R76E-expressing ARPE-19 cells was significantly less pronounced compared to their control (Figure 3B).

We also examined the levels of sqstm1/p62, a protein involved in autophagy, which can interact with the mitochondrial function. Like ferritin, the levels of p62 were slightly lower in control stable EPO-R76E cells compared to control ARPE-19 cells. Following 400 µM paraquat treatment, p62 levels increased in both cell lines. In ARPE-19 cells, this increase was approximately twofold compared to their control counterpart. In contrast, EPO-R76E-expressing cells showed only a minor increase in p62 levels after paraquat treatment compared to their control (Figure 3C).

**Figure 2 cells-14-01794-f002:**
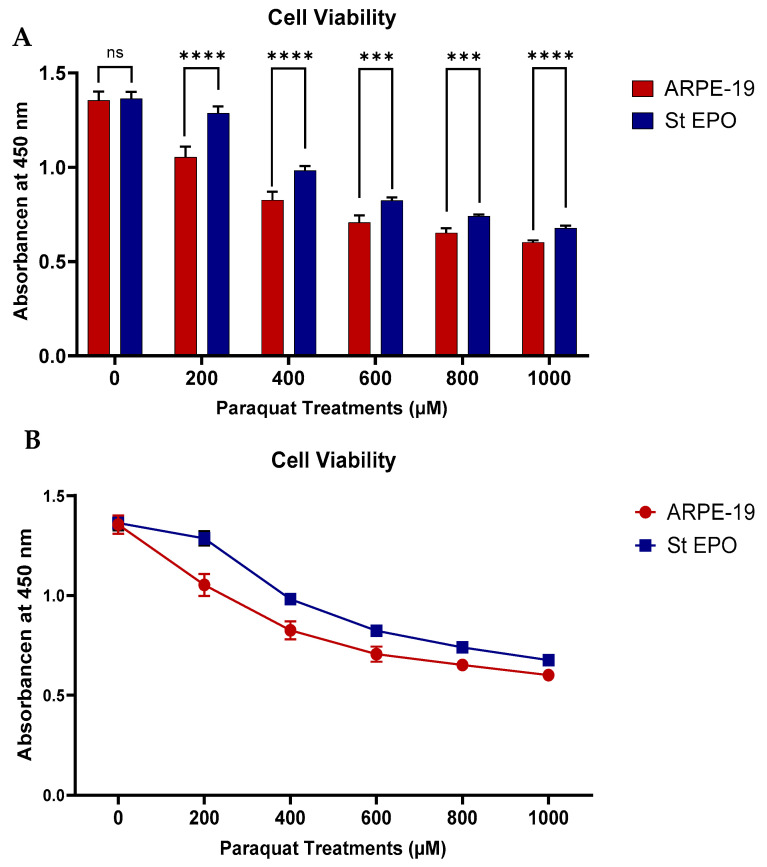
Cell viability of ARPE-19 cells under paraquat toxicity. (**A**) Bar graph and (**B**) line graph showing absorbance at 450 nm as a measure of cell viability in wild-type ARPE-19 cells (blue) and stable LentiEPOR-76E ARPE-19 cells (red) following treatment with increasing concentrations of paraquat (200–1000 µM). Data represent mean ± SD (*n* = 5). ns, *p* ≥ 0.05; *** *p* < 0.001; **** *p* < 0.0001.

Glutathione peroxidase 4 (GPX4), an important antioxidant enzyme that protects cells from oxidative stress and lipid peroxidation, was also assessed [17]. Under control conditions, GPX4 levels were slightly higher in stable EPO-R76E ARPE-19 cells compared to ARPE-19 control cells. Upon paraquat treatment, GPX4 levels increased in both cell types. The increase in ARPE-19 cells was slight compared to their control. Notably, EPO-R76E-expressing ARPE-19 cells exhibited a more substantial increase in GPX4 levels, approximately 1.5-fold higher compared to their control (Figure 3D).

Under basal conditions, NRF2 expression was slightly elevated in stable EPO-R76E-expressing ARPE-19 cells relative to control ARPE-19 cells. Upon exposure to 400 µM paraquat, NRF2 levels increased in both cell lines. In ARPE-19 cells, paraquat treatment led to a robust ~2-fold increase in NRF2 expression compared to untreated controls. In contrast, EPO-R76E-expressing cells exhibited only a modest increase in NRF2 levels post-treatment (Figure 3E).

**Figure 3 cells-14-01794-f003:**
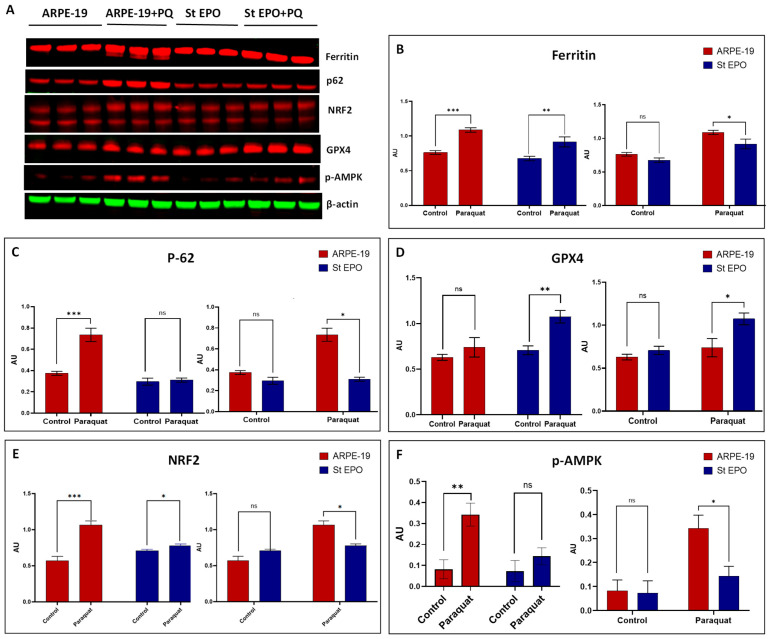
Western blot analysis of protein expression in ARPE-19 and stable EPO-R76E ARPE-19 cells following oxidative stress induction. (**A**) Representative Western blot images showing expression levels of ferritin (~21 kDa), p62/SQSTM1 (~48 kDa), GPX4 (~22 kDa), NRF2 (~66 kDa), and phosphorylated AMPK (p-AMPK; ~62 kDa) under basal (control) and oxidative stress conditions (400 µM paraquat for 24 h) in ARPE-19 and EPO-R76E-expressing ARPE-19 cells. β-actin (~42 kDa) was used as a loading control and detected in green, while target proteins were visualized in red. (**B**–**F**) Quantification of Western blot band intensities normalized to β-actin for each protein: (**B**) Ferritin expression under control and paraquat-treated conditions in both cell lines. (**C**) p62 expression comparison across treatment groups and cell lines. (**D**) GPX4 levels demonstrating the impact of EPO-R76E on antioxidant defense under stress. (**E**) NRF2 expression illustrating activation under oxidative stress. (**F**) p-AMPK activation profile across conditions. Data are presented as mean ± SD (*n* = 3). ns: *p* ≥ 0.05, * *p* < 0.05, ** *p* < 0.01, *** *p* < 0.001.

Phosphorylated AMP-activated protein kinase (p-AMPK), a stress-sensing kinase, showed increased levels in both ARPE-19 and stable EPO-R76E ARPE-19 cells after 400 µM paraquat treatment. In ARPE-19 cells, the increase in p-AMPK was more than three times the control level. In EPO-R76E-expressing cells, the p-AMPK level approximately doubled compared to their control (Figure 3F).

### 3.3. Impact of EPO-R76E on Intracellular Ferrous Ion Content

To investigate the effect of EPO-R76E expression on intracellular iron accumulation, fluorescent imaging was performed (Figure 4A), and signal intensity was quantified (Figure 4B). In ARPE-19 cells, paraquat treatment led to a marked increase in FerroOrange intensity, indicating substantial accumulation of intracellular ferrous ions. In contrast, EPO-R76E-expressing ARPE-19 cells exhibited significantly lower fluorescence intensity under identical treatment conditions, suggesting reduced Fe^2+^ accumulation. Moreover, under control conditions (no paraquat), both ARPE-19 and EPO-R76E-expressing cells showed minimal basal FerroOrange fluorescence (Figure 4B).

### 3.4. Impact of EPO-R76E on Reactive Oxygen Species Content

To evaluate intracellular reactive oxygen species (ROS) levels during paraquat-induced oxidative stress, fluorescence intensity was quantified. Paraquat treatment led to a significant increase in DCFDA fluorescence, indicating elevated ROS production compared to the untreated control group (Figure 5A). Quantitative analysis revealed that the ROS level almost doubled for paraquat-induced oxidative stressed ARPE-19 cells, while EPO-R76E expressing oxidative stressed ARPE-19 cells showed significantly lower ROS levels (approximately 1.3-fold compared to control) (Figure 5B).

## 4. Discussion

The findings demonstrate that stable expression of EPO-R76E effectively protects ARPE-19 cells from oxidative stress-induced cell death, a process significantly implicated in the pathogenesis of age-related macular degeneration (AMD).

The cell viability assay clearly shows that EPO-R76E-expressing cells maintain considerably higher viability across increasing concentrations of the oxidative stress inducer, paraquat, compared to control cells (Figure 2). Stable LentiEPOR-76E ARPE-19 cells exhibit higher absorbance value compared to ARPE-19 cells, suggesting its protective effect against oxidative stress. This protective effect aligns with previous research highlighting EPO-R76E’s ability to inhibit oxidative stress in retinal degeneration models [12].

To further dissect how EPO-R76E counteracts oxidative stress, we examined the expression of key proteins involved in this cell death pathway. Ferritin, an iron storage protein, often rises in response to iron overload and oxidative stress associated with ferroptosis [18]. Our results demonstrate that ferritin levels, a key indicator of cellular iron storage and iron-mediated oxidative stress, were significantly elevated in response to paraquat treatment in ARPE-19 cells (Figure 3B). This is consistent with previous studies showing that oxidative agents like paraquat increase intracellular labile iron pools and upregulate ferritin expression as part of the cellular defense mechanism against iron-induced reactive oxygen species (ROS) and ferroptosis [11,19]. Ferritin functions by sequestering excess iron, thereby limiting the Fenton reaction and subsequent lipid peroxidation that are characteristic of ferroptotic cell death [20]. Notably, EPO-R76E-expressing ARPE-19 cells exhibited a blunted ferritin response to paraquat exposure when compared to their ARPE-19 counterparts. This suggests that EPO-R76E may modulate iron homeostasis and reduce intracellular oxidative stress, either by suppressing labile iron release or enhancing antioxidant defenses.

In addition to examining ferritin, we evaluated the expression of p62/SQSTM1, an autophagy adaptor protein that also plays a regulatory role in mitochondrial function and oxidative stress pathways [5,17]. Our data shows that p62 levels were modestly reduced in EPO-R76E-expressing cells under basal conditions compared to parental ARPE-19 cells (Figure 3C). Upon treatment with 400 µM paraquat, both cell types exhibited an increase in p62 expression; however, the magnitude of induction differed significantly. While ARPE-19 cells showed an approximate twofold increase, the response in EPO-R76E-expressing cells was noticeably attenuated.

This differential expression pattern of p62 suggests distinct stress-handling strategies between the two cell types. Under basal conditions, the slightly reduced levels of p62 in EPO-R76E-expressing cells may reflect enhanced basal autophagic flux, consistent with a pre-adapted state of mitochondrial quality control and protein homeostasis. Since p62 is degraded during autophagy, lower levels can indicate increased autophagic activity [21]—a protective mechanism potentially promoted by EPO-R76E signaling.

Following oxidative insult, the increase in p62 in ARPE-19 cells may reflect a stress-induced accumulation of autophagic cargo or a compensatory upregulation of selective autophagy. In contrast, the blunted response in EPO-R76E-expressing cells could be indicative of more efficient basal clearance mechanisms or a lower accumulation of damaged proteins and organelles, reducing the need for p62 upregulation. Additionally, p62 is known to participate in the positive regulation of NRF2 through competitive binding with Keap1, linking oxidative stress and autophagy signaling pathways [22]. The modest increase in both p62 and NRF2 (Figure 3E) in EPO-R76E cells after paraquat treatment may reflect a tightly regulated feedback loop maintaining redox and proteostatic balance. Moreover, emerging evidence suggests that p62 also intersects with ferroptosis [23]. The attenuation of p62 upregulation in EPO-R76E-expressing cells could imply a protective role of EPO-R76E against oxidative stress, possibly through the modulation of autophagic degradation pathways or antioxidant defense systems.

We investigated the impact of EPO-R76E expression on GPX4 levels in ARPE-19 cells under oxidative stress induced by paraquat. Our data reveals that EPO-R76E-expressing cells maintain slightly elevated basal GPX4 expression relative to control ARPE-19 cells. More notably, following treatment with paraquat, GPX4 expression increased significantly in EPO-R76E cells, suggesting a heightened adaptive antioxidant response (Figure 3D). GPX4 is a selenoenzyme essential for detoxifying lipid peroxides, and its activity is considered a critical defense mechanism against oxidative stress [24]. The observed upregulation of GPX4 aligns with the protective role of EPO-R76E and implies a mechanistic link between EPO-R76E signaling and antioxidant response. It is plausible that EPO-R76E enhances GPX4 expression via Nrf2 stabilization or nuclear translocation under oxidative stress conditions. This hypothesis is supported by recent findings demonstrating that Nrf2 activation mitigates ferroptotic cell death through upregulation of antioxidant enzymes, including GPX4 and SLC7A11 [25]. The increased GPX4 expression in paraquat-treated EPO-R76E cells may also reflect feedback adaptation to elevated intracellular ROS. The preservation of GPX4 under such conditions is crucial, as its depletion is a defining event in oxidative stress inhibition. Therefore, the EPO-R76E-mediated enhancement of GPX4 provides a compelling explanation for the improved resistance to oxidative injury observed in these cells. Our future studies will include ACSL4, SLC7A11, and lipid peroxidation assays to fully dissect ferroptotic versus apoptotic contributions.

To assess the role of EPO-R76E in modulating antioxidant responses, we examined the protein expression levels of NRF2, a key transcription factor regulating cellular antioxidant defense and mitochondrial quality control. Our findings highlight a modulatory effect of EPO-R76E on the basal and inducible levels of NRF2 under oxidative stress conditions.

At baseline, NRF2 expression was modestly elevated in EPO-R76E-expressing cells compared to parental ARPE-19 cells, suggesting that EPO-R76E may promote a pre-activated antioxidant state (Figure 3E). Upon treatment with 400 µM paraquat, which generates reactive oxygen species and impairs mitochondrial function, NRF2 levels increased in both cell lines. Notably, parental ARPE-19 cells showed an approximate twofold induction, while the EPO-R76E-expressing cells exhibited only a minor increase, indicating a dampened stress response.

A particularly interesting observation was the appearance of NRF2 as a double band in Western blot analysis. This pattern is consistent with previously reported post-translational modifications of NRF2, most notably phosphorylation, which can alter NRF2’s electrophoretic mobility and functional state [26]. The upper band is typically associated with the phosphorylated or activated form of NRF2, while the lower band may represent the unmodified or partially degraded protein [27]. The presence of both bands suggests dynamic regulation of NRF2 stability and activity in response to oxidative cues.

Importantly, the reduced induction of NRF2 in EPO-R76E cells upon paraquat exposure, along with the persistence of the upper band at baseline, may reflect a state of NRF2 priming or reduced ROS accumulation due to enhanced mitochondrial resilience. This could be attributed to EPO-R76E’s known effects on promoting mitochondrial quality control and reducing oxidative burden through pathways such as PI3K/Akt and STAT5 [28]. Thus, the blunted NRF2 response upon stress in these cells may not signify impaired defense, but rather an already optimized antioxidant status requiring less acute activation. These findings underscore the potential of EPO-R76E to modulate cellular redox homeostasis and precondition retinal cells against oxidative injury. Further work examining HO-1 induction and Nrf2 nuclear translocation will strengthen the mechanistic understanding in future studies.

Another of the key signaling molecules evaluated was phosphorylated AMP-activated protein kinase (p-AMPK), a critical regulator of energy homeostasis. AMPK activation serves as a cellular mechanism to maintain metabolic homeostasis and promote cell survival when mitochondria are impaired [29]. AMPK is known to promote mitochondrial biogenesis and oxidative balance under physiological stress [30]. Interestingly, expression of EPO-R76E was associated with a reduction in p-AMPK levels and mitigation of oxidative stress markers, suggesting that EPO-R76E may preserve mitochondrial integrity by modulating AMPK activity (Figure 3F). Our findings align with prior reports indicating that AMPK overexpression can disrupt mitochondrial dynamics by promoting fission, suppressing fusion, and impairing oxidative phosphorylation, which indicates the accumulation of dysfunctional mitochondria and increased ROS production [31]. In our paraquat-treated ARPE-19 cells, elevated p-AMPK levels coincided with increased ROS and mitochondrial damage, consistent with the notion that hyperactivation of AMPK may exacerbate oxidative stress. This is particularly relevant in the context of AMD, where mitochondrial damage and oxidative stress are critical drivers of RPE degeneration and disease progression. In our study, EPO-R76E–expressing cells show increased Nrf2 protein levels along with reduced AMPK activation, supporting a model in which EPO-R76E mitigates oxidative stress and allows coordinated activation of Nrf2-dependent cytoprotective pathways. While our study demonstrates that EPO-R76E reduces oxidative stress and modulates AMPK and Nrf2 signaling, we acknowledge that further analyses of downstream targets are required. Nevertheless, these results offer a valuable mechanistic insight for future studies investigating the pathways underlying EPO-R76E–mediated cytoprotection.

In this study, we investigated the protective effects of EPO-R76E against oxidative stress in ARPE-19 cells and found that its expression was associated with reduced intracellular ROS and ferrous ion levels. Excess intracellular iron causes oxidative stress by catalyzing the production of reactive oxygen species (ROS) through reactions like the Fenton reaction, and ROS content is an identifying factor for oxidative stress in RPE cells. Our data show that paraquat exposure markedly increases both intracellular ferrous iron and reactive oxygen species (ROS) levels in ARPE-19 cells, as detected by FerroOrange and DCFDA assays, respectively. These findings align with previous reports that paraquat induces oxidative damage by promoting mitochondrial dysfunction and redox cycling, thereby elevating ROS and labile iron pools [19,32]. Interestingly, the basal level of intracellular ferrous ion content in EPO-R76E-expressing ARPE-19 cells was slightly higher than that observed in ARPE-19 cells under control conditions; however, this difference was not statistically significant. This modest elevation may reflect cell line variability or altered iron handling associated with EPO-R76E expression, though it does not appear to impact cellular redox status in the absence of oxidative stress. Moreover, the oxidative stress-induced EPO-R76E-expressing ARPE-19 cells demonstrated significantly reduced levels of intracellular Fe^2+^ (Figure 4B) and ROS (Figure 5B) following paraquat treatment compared to the ARPE-19 cells, suggesting that EPO-R76E confers protection against mitochondrial oxidative damage. Notably, the intracellular ferrous ion content in paraquat-treated EPO-R76E-expressing ARPE-19 cells was comparable to or slightly lower than that of untreated EPO-R76E cells. This observation is somewhat unexpected, as oxidative stress typically elevates labile iron pools by promoting ferritin degradation or mitochondrial iron release [33]. The result suggests that EPO-R76E expression may buffer iron homeostasis even under pro-oxidant conditions. However, further investigation is needed to confirm this result. Overall, the reduced FerroOrange and DCFDA fluorescence observed in EPO-R76E cells indicates a more robust antioxidant defense and diminished oxidative stress. Future studies will incorporate confocal and FACS-based analyses to further substantiate these findings.

## 5. Conclusions

Taken together, these results indicate that EPO-R76E protects RPE cells from oxidative stress-induced cell death. It reduces overall mitochondrial damage from ROS and iron accumulation and appears to modulate key components of the retinal degeneration. The interplay between p62, NRF2, and mitochondrial quality control suggests that EPO-R76E fosters a preconditioned cellular state with enhanced resilience to oxidative injury. The enhanced expression of GPX4 in EPO-R76E-expressing cells under stress conditions stands out as a particularly significant factor in preserving cell viability. These findings support the potential of EPO-R76E as a therapeutic candidate for AMD by targeting RPE oxidative stress-induced cell death. Future studies should aim to investigate autophagic flux directly using lysosomal inhibitors and assess autophagy-specific markers to further elucidate the mechanistic links between EPO-R76E signaling, p62 regulation, and stress adaptation in retinal pigment epithelial cells.

## Figures and Tables

**Figure 1 cells-14-01794-f001:**
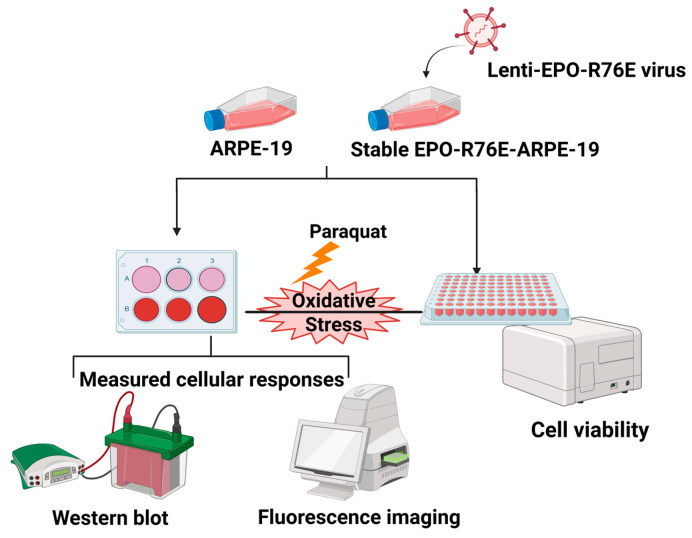
Schematic representation of experimental design. ARPE-19 cells were transduced with LentiEPO-R76E virus using polybrene, and stable clones were selected using puromycin. Oxidative stress was induced by treating cells with 400 μM paraquat for 24 h. Subsequent assays included intracellular ROS measurement using DCFDA, ferrous ion quantification using FerroOrange, and Western blot analysis of oxidative stress-associated markers. https://www.bioRender.com (accessed on 5 November 2025).

**Figure 4 cells-14-01794-f004:**
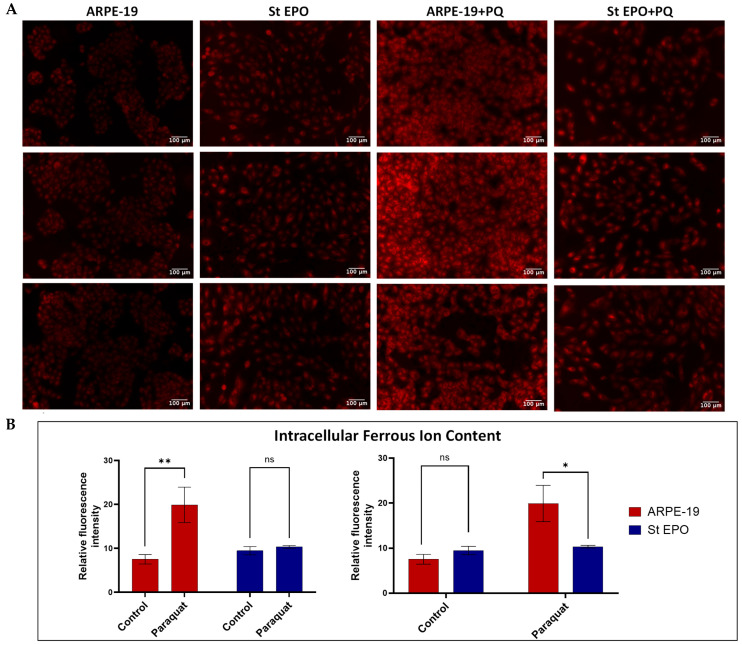
Intracellular Ferrous Ion (Fe^2+^) Levels Assessed by FerroOrange Staining. (**A**) Fluorescence images of ARPE-19 cells stained with FerroOrange, a selective fluorescent probe for labile Fe^2+^. Red fluorescence corresponds to intracellular ferrous ion content. Cells were treated under indicated conditions and imaged using fluorescence microscopy. (**B**) Quantification of FerroOrange fluorescence intensity from the corresponding images using ImageJ. Fluorescence intensity correlates with intracellular Fe^2+^ accumulation. Data are presented as mean ± SD (*n* = 3). Scale bar: 100 µm. ns: *p* ≥ 0.05, * *p* < 0.05, ** *p* < 0.01.

**Figure 5 cells-14-01794-f005:**
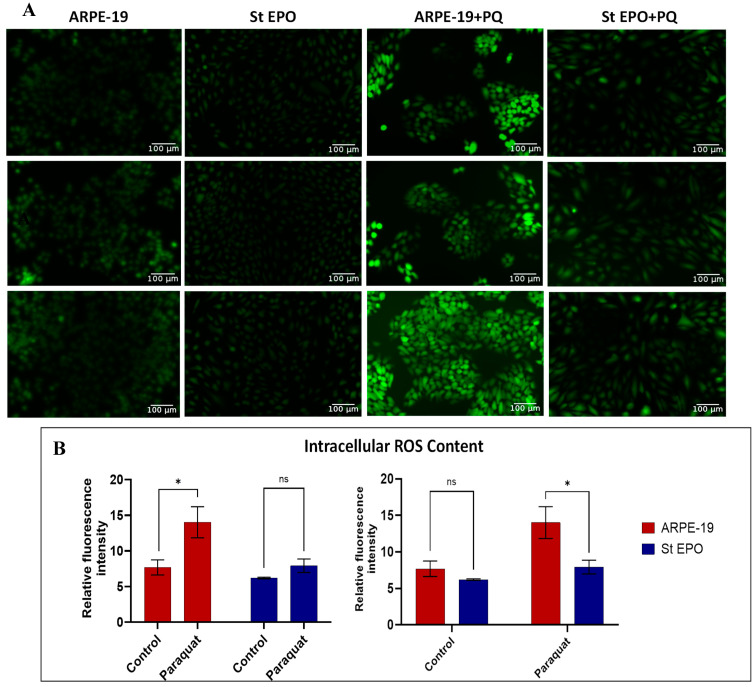
ROS Levels Assessed by DCFDA Staining. (**A**) Fluorescence images of ARPE-19 cells stained with 2′,7′-dichlorofluorescin diacetate (DCFDA), a cell-permeable indicator for ROS. Green fluorescence corresponds to ROS content. Cells were treated under the indicated conditions, and green fluorescence (oxidized DCF) was visualized using fluorescence microscopy. (**B**) Quantification of fluorescence intensity from the corresponding images using ImageJ, indicating relative intracellular ROS levels. Data are presented as mean ± SD (*n* = 3). Scale bar: 100 µm. ns: *p* ≥ 0.05, * *p* < 0.05.

## Data Availability

The original contributions presented in this study are included in the article/Appendix A. Further inquiries can be directed to the corresponding author.

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
