# Peer review of "EPO-R76E Enhances Retinal Pigment Epithelium Viability Under Mitochondrial Oxidative Stress Induced by Paraquat"

_cells, 2025, doi:10.3390/cells14221794_

Round 1
Reviewer 1 Report
Comments and Suggestions for Authors
In this manuscript, the authors investigate the cytoprotective role of the non-erythropoietic EPO variant, EPO-R76E, in suppressing oxidative stress in RPE cells. While the topic is of potential interest, the data presented do not provide sufficient experimental support for the conclusions.
Major Concerns:
- Figures 1–2: The schematic and cloning strategy are not essential to the main message and could be moved to the supplementary material. Methodological details should be provided in the Methods section.
- Cell viability: The level of protection observed is modest (<30% at 200 µM paraquat, and minimal at higher concentrations). At 400 µM paraquat, the defense in transfected cells is still small. Given this, it is difficult to expect robust downstream effects or mechanistic insights.
- Choice of paraquat dose: The rationale for using 400 µM paraquat is unclear, especially since the rescue effect is small.
- Fe+ and ROS detection: The fluorescence microscopy approach is insufficiently described (e.g., which imaging system was used). The image quality is low, and the conclusions would be stronger if supported by confocal microscopy for Fe+ or flow cytometry (FACS) for ROS determination.
- Nrf2 signaling: The measurement of Nrf2 alone is not an adequate marker of pathway activation. Additional analyses, such as Keap1 expression, HO-1 induction, and/or nuclear translocation of Nrf2, would provide a more convincing demonstration.
- AMPK signaling: The role of AMPK in this context is not clear from the data provided. Further clarification or additional experiments would be required.
Overall assessment:
The study does not present a sufficiently coherent or convincing mechanistic story. One of the main limitations is the relatively small degree of cytoprotection achieved (Fig. 4), which weakens the interpretation of downstream effects.
Reviewer 2 Report
Comments and Suggestions for Authors
The manuscript addresses a highly relevant and timely topic in the field of age-related macular degeneration (AMD), investigating the potential of the non-erythropoietic variant EPO-R76E as a protective agent for retinal pigment epithelium (RPE) against oxidative stress. The study is well-conducted, with robust validation of the generated stable cell line and consistent results supporting the protective role of EPO-R76E.
However, there are some critical issues that limit the impact of the work, including the restricted model used, the lack of direct mitochondrial functional assays, and the absence of comparison with wild-type EPO. Addressing these would significantly strengthen the manuscript.
Major comments
- The study relies exclusively on ARPE-19 cells, which differ significantly from native RPE in terms of metabolism and oxidative stress response. Validation in primary RPE cultures or an in vivo model would considerably enhance the translational value of the findings.
- Although the authors hypothesize that EPO-R76E preserves mitochondrial integrity, no direct assays were performed (e.g., mitochondrial membrane potential, OCR/ECAR with Seahorse analysis, intracellular ATP levels). Including such measurements would strengthen the mechanistic conclusions.
- The novelty of EPO-R76E lies in its non-erythropoietic activity, yet the manuscript does not provide a side-by-side comparison with wild-type EPO. Demonstrating whether EPO-R76E performs better, worse, or equivalently in protecting against paraquat toxicity would considerably increase the impact of the work.
- The protein panel examined (ferritin, p62, NRF2, p-AMPK, GPX4) is informative but not exhaustive. Additional ferroptosis markers (e.g., ACSL4, SLC7A11, lipid peroxidation assays) and apoptotic/necrotic markers would help clarify the specific pathways involved in the protective effect of EPO-R76E.
- The manuscript briefly suggests the therapeutic potential of EPO-R76E in AMD but does not discuss the challenges of gene therapy delivery, safety, or off-target effects. Expanding this section would provide a more balanced and clinically meaningful perspective.
Minor comments
- Some microscopy images (FerroOrange, DCFDA) appear low in resolution. Higher-quality images and more detailed quantifications (e.g., time-course analyses) would improve clarity.
- Inconsistency is observed in how results are reported: some experiments show raw absorbance values (e.g., WST-1), while others show normalized data. A uniform approach would improve readability.
- The rationale for using paraquat as an oxidative stress inducer should be expanded, explaining its specific relevance to AMD-related mitochondrial dysfunction.
- Some sentences, particularly in the Materials and Methods section, are overly long and redundant. Editing for conciseness would enhance clarity.
- The discussion would benefit from integrating recent literature (last 2–3 years) on ferroptosis and AMD to better contextualize the findings.
This is a promising and well-structured study, but additional functional experiments and a more critical discussion are required before publication.
Recommendation: Major revision
Round 2
Reviewer 1 Report
Comments and Suggestions for Authors
The authors have responded appropriately to the reviewers’ comments. Although the study’s impact is limited, the work is technically sound, and I have no further concern
Reviewer 2 Report
Comments and Suggestions for Authors
Accepted in this form